# Association of Midgut Bacteria and Their Metabolic Pathways with Zika Infection and Insecticide Resistance in Colombian *Aedes aegypti* Populations

**DOI:** 10.3390/v14102197

**Published:** 2022-10-06

**Authors:** Andrea Arévalo-Cortés, Ashish Damania, Yurany Granada, Sara Zuluaga, Rojelio Mejia, Omar Triana-Chavez

**Affiliations:** 1Group Biología y Control de Enfermedades Infecciosas, Universidad de Antioquia—UdeA, Calle 70 No. 52-21, Medellín 050010, Colombia; 2National School of Tropical Medicine, Baylor College of Medicine, Houston, TX 77030, USA; 3Program for Innovative Microbiome and Translational Research, Department of Genomic Medicine, The University of Texas MD Anderson Cancer Center, Houston, TX 77030, USA

**Keywords:** *Aedes aegypti*, Zika, lambda-cyhalothrin, bacteria, midgut, dysbiosis, functions

## Abstract

Introduction: *Aedes aegypti* is the vector of several arboviruses such as dengue, Zika, and chikungunya. In 2015–16, Zika virus (ZIKV) had an outbreak in South America associated with prenatal microcephaly and Guillain-Barré syndrome. This mosquito’s viral transmission is influenced by microbiota abundance and diversity and its interactions with the vector. The conditions of cocirculation of these three arboviruses, failure in vector control due to insecticide resistance, limitations in dengue management during the COVID-19 pandemic, and lack of effective treatment or vaccines make it necessary to identify changes in mosquito midgut bacterial composition and predict its functions through the infection. Its study is fundamental because it generates knowledge for surveillance of transmission and the risk of outbreaks of these diseases at the local level. Methods: Midgut bacterial compositions of females of Colombian *Ae. aegypti* populations were analyzed using DADA2 Pipeline, and their functions were predicted with PICRUSt2 analysis. These analyses were done under the condition of natural ZIKV infection and resistance to lambda–cyhalothrin, alone and in combination. One-step RT-PCR determined the percentage of ZIKV-infected females. We also measured the susceptibility to the pyrethroid lambda–cyhalothrin and evaluated the presence of the V1016I mutation in the sodium channel gene. Results: We found high ZIKV infection rates in *Ae. aegypti* females from Colombian rural municipalities with deficient water supply, such as Honda with 63.6%. In the face of natural infection with an arbovirus such as Zika, the diversity between an infective and non-infective form was significantly different. Bacteria associated with a state of infection with ZIKV and lambda–cyhalothrin resistance were detected, such as the genus *Bacteroides*, which was related to functions of pathogenicity, antimicrobial resistance, and bioremediation of insecticides. We hypothesize that it is a vehicle for virus entry, as it is in human intestinal infections. On the other hand, Bello, the only mosquito population classified as susceptible to lambda–cyhalothrin, was associated with bacteria related to mucin degradation functions in the intestine, belonging to the *Lachnospiraceae* family, with the genus *Dorea* being increased in ZIKV-infected females. The *Serratia* genus presented significantly decreased functions related to phenazine production, potentially associated with infection control, and control mechanism functions for host defense and quorum sensing. Additionally, *Pseudomonas* was the genus principally associated with functions of the degradation of insecticides related to tryptophan metabolism, ABC transporters with a two-component system, efflux pumps, and alginate synthesis. Conclusions: Microbiota composition may be modulated by ZIKV infection and insecticide resistance in *Ae. aegypti* Colombian populations. The condition of resistance to lambda–cyhalothrin could be inducing a phenome of dysbiosis in field *Ae. aegypti* affecting the transmission of arboviruses.

## 1. Introduction

Mosquitoes such as *Aedes aegypti* are a global public health problem since they transmit infectious agents such as dengue virus (DENV), Zika virus (ZIKV), and Chikungunya virus (CHIKV) that cause diseases for which there are no effective antiviral therapies or vaccines [1]. Therefore, efforts to cut transmission have been focused on mosquito vector control using insecticides, but this strategy has problems associated with developing resistance in mosquito populations [2,3,4,5].

Zika fever was considered a neglected tropical disease before 2013 because it had only local outbreaks [6,7]. ZIKV belongs to the *Flaviviridae* family and is transmitted principally by *Ae. aegypti* [8,9,10]. This arbovirus had a significant outbreak between 2015–2016 in Central and South America, associated with a high incidence of neurological abnormalities, prenatal microcephaly, and Guillain-Barré syndrome [11,12]. Although it is not clear if ZIKV outbreaks impacted subsequent DENV transmission, in 2017, there was a generalized decreased incidence of DENV and ZIKV with a record low in many countries in the Americas, with a resurgence in dengue cases occurring, such as the one seen in Brazil in 2019 [13,14].

In Colombia, at the end of 2015, an increase in ZIKV cases was observed according to the public health surveillance system, and the epidemic reached its highest number with 6386 cases in epidemiological week 05 of 2016 [15]. During the epidemic phase, the incidence of ZIKV was 377.7 cases per 100,000 inhabitants in the urban population [16]. In 2017, 2130 cases were notified, marking a 97.6% decrease in cases compared to 2016 [17]. Between 2018 and 2019, a clear drop in cases was observed, reaching 1.6 cases per 100,000 inhabitants in the last week of 2019 [18,19].

Additionally, *Ae. aegypti* presents symbiotic relationships with its associated microbiota (bacteria, fungi, viruses, etc.) [20]. The importance of bacterial composition, abundance, and diversity in mosquitoes and their interactions with the vector have been highlighted by their influence on susceptibility to and transmission of pathogens [21,22,23,24,25,26,27].

Gut bacteria might directly or indirectly mediate anti-viral/anti-parasitic or pro-viral/pro-parasitic mechanisms in the vector mosquitoes [24,28,29,30]. The influence of *Wolbachia* in *Ae. aegypti* is an example of anti-viral activity against infection by DENV, ZIKV, and CHIKV, as this bacterium modulates the mosquito’s immune response (Toll, IMD, and microRNAs pathways), induces reactive oxygen species (ROS), and triggers the production of antimicrobial peptides (AMPs) [29,31,32,33,34,35,36,37].

Furthermore, *Wolbachia* has an anti-parasitic effect by controlling *Plasmodium gallinaceum, P. berghei,* and *P. falciparum* infection [29,38,39], as it stimulates immune genes expression and induces the production of ROS in *Anopheles gambiae* and *An. stephensi,* respectively [38,39]. Other bacteria that show anti-pathogenic properties are *Chromobacterium* and *Proteus*; the first reduces susceptibility to DENV and *P. falciparum* infections, while the second only to DENV infection [23,30].

However, in *Ae. aegypti*, bacteria such as *Serratia marcescens,* and *S. odorifera* promote permissiveness to arboviruses [24,40,41]. The protein *Sm*Enhancin secreted by *S. marcescens* digests mosquito gut membrane-bound mucins, enhancing viral dissemination of DENV, ZIKV, and Sindbis virus (SINV) [24]. Moreover, the polypeptide P40 secreted by *S. odorifera* interferes with virus recognition by the mosquito midgut surface proteins prohibitin and porin, increasing *Ae. aegypti* permissiveness to DENV and CHIKV, respectively [40,41].

These studies highlight the importance of understanding the metabolites and functions carried out by bacteria since components derived from them may benefit the vector but simultaneously be antagonists and help establish infection [28,42]. This is the case with *Wigglesworthia*, which is a mutualistic symbiont of the tsetse fly *Glossina* spp. that provides folate (vitamin B9), which is fundamental for fly reproduction; however, the presence of this nutrient is exploited by *Trypanosoma brucei* to successfully establish infection in the midgut and complete its life cycle [42].

Additionally, bacteria have been related to other characteristics of mosquito physiology, such as development, nutrition, reproduction, and resistance to insecticides [43,44,45,46,47]. In this last aspect, it has been shown in *Ae. aegypti* that midgut bacterial communities participate in detoxifying insecticides such as propoxur carbamate and naled organophosphate [48], and we recently described bacterial signatures associated with lambda–cyhalothrin resistance that could play an essential role in the metabolism of pyrethroid insecticides at higher concentrations [47]. Likewise, midgut bacterial communities were altered after laboratory selection of *Ae. aegypti* and *Culex pipiens quinquefasciatus* by permethrin and deltamethrin, respectively, with a higher abundance of *Cutibacterium*, *Leucobacter*, *Rickettsia*, *Bacillus*, *Pseudomonas*, and *Wolbachia* in insecticide-selected strains. Additionally, these associated bacteria have been potentially related to the degradation of chemical compounds and activation of enzymes that detoxify insecticides, such as glutathione S-transferases (GSTs) and cytochrome P450s (CYP450s) [49,50].

This work aimed to identify changes in the bacterial composition associated with midgut *Ae. aegypti* with natural ZIKV infections and to predict its functions through PICRUSt2 analysis under natural infection conditions and lambda–cyhalothrin resistance. Understanding bacterial metabolic functions associated with vector insects can provide important information on candidate pathways or metabolites that play a role in infection and resistance processes, offering a range of possibilities that could be used in the future to intervene or alter innovative control strategies for these mosquito-borne diseases.

## 2. Materials and Methods

### 2.1. Mosquito Collection

Female mosquitoes were collected between 2016 and early 2017 using sweep nets in six Colombian cities in 10 to 20 randomized houses from four neighborhoods in each city. Staff involved in vector-borne disease programs from each municipality assisted mosquito collection. The location of samples analyzed in this study is shown in Figure 1. These cities were selected because they comprise the majority of arbovirus cases in Colombia [16,19,51,52,53]. Some have had a significant epidemiological history for dengue in the last ten years [52,53]. In addition, we chose rural areas where entomological surveillance is deficient. The distance between the closest cities (Honda and Puerto Bogotá) is approximately 7 km. In Colombia, it has been shown that even in geographically close populations there may be differences for *Ae. aegypti* in vector competence for DENV [54,55] and in hatching, development time, longevity, and fecundity [56]. This suggests that some specific parameters at the local level, such as bacterial diversity or resistance to insecticides, may be differential and relevant to elucidate the different degrees of transmission of arboviruses.

F0 field females were instantly placed in individual tubes and maintained on ice until subsequent taxonomic identification was made. Mosquitoes were identified by the patterns of typical white lyre-shaped markings on thoracic scales [57]. Immediately after identification, mosquitoes were stored individually at −80 °C until DNA extraction from the midgut was performed for DNA 16S sequencing.

### 2.2. ZIKV detection in Field Females

Female carcasses and other organs, excluding the midgut, were macerated mechanically following the protocol recommended by QIAGEN. Total RNA was extracted using the RNeasy Mini Kit^®^ (QIAGEN Germantown, MD, USA, Catalog No. 74104). To detect females infected with ZIKV, one-step RT-PCR was performed with the Luna^®^ Universal Probe One-Step RT-qPCR kit (New England BioLabs Inc. Ipswich, MA, USA, Catalog No. E3006S). A volume of 10μL reaction contained: 1 μL of RNA, 1× enzyme mix, 1× reaction mix, and 0.4 μM of each of the primers ZIKVF9027-5’-CCTTGGATTCTTGAACGAGGA-3’ and ZIKVR9197-5’-AGAGCTTCATTCTCCAGATCAA-3’, which amplified a specific 192 bp region of the NS5 gene of the ZIKV [58]. The thermal profile for the RT-PCR was as follows: reverse transcription at 55 °C for 10 min, initial denaturation at 95 °C for 1 min, followed by 40 amplification cycles at 95 °C for 10 s, 56 °C for the 30 s, and 72 °C for 20 s. Negative controls for extraction and amplification were included in each reaction, and ZIKV RNA from the supernatant of infected C6/36HT cells was included as a positive control. Negative and positive controls were processed simultaneously and under the same conditions as the samples. The amplification products were analyzed on 2.5% agarose gels in 0.5× Tris Borate EDTA (TBE), stained with ethidium bromide (0.6 μg/mL), and visualized under (ultraviolet) UV light. Significant differences between cities according to infection status were established using the Chi-squared (*Χ^2^*) or Fisher’s exact tests (when the expected value was <5) for count data.

### 2.3. Determination of lambda–cyhalothrin Resistance Profile and Allele-Specific PCR (AS-PCR) for the kdr Mutation V1016I

The resistance status for lambda–cyhalothrin in each *Ae. aegypti* population was determined using sixty F1 larvae from the third or fourth instar following the standardized methodology of the World Health Organization (WHO) [59], which is explained in more detail in our previous work [47]. Larvae were exposed to one of six concentrations of lambda–cyhalothrin (0.000468 ppm to 0.06 ppm) to determine larval mortality 24 h after exposure. Although pyrethroids are not used for larval treatment, we tested them against *Ae. aegypti* to obtain information on the larval resistance status that may reflect the adult resistance status since the target of pyrethroids is a constitutively expressed gene. A resistance ratio (RR) was obtained by dividing the LC50 of each population by the equivalent LC50 of the Rockefeller reference susceptible strain. The RR was interpreted as being susceptible to insecticide if <5-fold, as well as having moderate insecticide resistance (5- to 10-fold) or high insecticide resistance (>10-fold) [47,59].

Additionally, the presence of the V1016I mutation was determined in F0 adults with an allele-specific PCR according to previously published methods [47], using the primers reported by Li et al., 2015 [60]: PM2_Ext_1016F GCCACCGTAGTGATAGGAAATC; PM2_Ext_1016R CGGGTTAAGTTTCGTTTAGTAGC, and by Granada et al., 2018 [4]: PM2_F_1016Wt GTTTCCCACTCGCACAGGT; PM2_F_1016Mut GTTTCCCACTCGCACAGA. A minimum of 30 adult mosquitoes from each city was processed to identify the V1016I mutation. The allele frequencies were determined by comparing the number of each of the alleles (wild and mutated) with the total number of alleles in the population using the Microsoft Excel program. The results of both tests were published in Arévalo-Cortés et al., 2020 [47].

### 2.4. Preparation of Genomic DNA and Library for Metagenome 16S Sequencing

Female mosquitoes were surface-sterilized by dipping them in 70% ethanol for 5 min and then rinsed twice for 1 min in sterile Dulbecco’s phosphate-buffered saline (DPBS) solution. The midgut from each mosquito was dissected under sterile conditions in a drop of sterile DPBS, taking care not to contaminate it with any other tissues, and total DNA was extracted from each midgut with the ZymoBIOMICS^TM^ DNA Miniprep kit (ZYMO RESEARCH, Irvine, CA, USA; Catalog No. D4300). Dissections and extractions were performed in an aseptic environment to avoid contamination. Additionally, a mock sample spiked with the ZymoBIOMICS^TM^ Microbial Community Standard (ZYMO RESEARCH, Irvine, CA, USA; Catalog No. D6300), which contains samples from 10 well-known species of bacteria, was used in all procedures as environmental contamination controls. The DNA was sent to Macrogen (Seoul, South Korea), where the libraries were prepared and sequenced. Briefly, DNA integrity was verified on a 2100 Bioanalyzer (Agilent Technologies, Santa Clara, CA, USA), and the prepared libraries were quantified using the Illumina qPCR Quantification Protocol Guide. The bacterial 16S V3–V4 region was sequenced for 63 DNA samples (62 from mosquito midguts and the mock sample) on an Illumina Miseq platform using the bacterial and archaeal universal primers 16S_V3-341F: (5′-CCTACGGGNGGCWGCAG-3′) and 16S_V4-785R: (5′-GACTACHVGGGTATCTAATCC-3′).

### 2.5. Microbial Community Analysis

Pre-processing of the fastq reads and taxonomic analysis was performed using DADA2 Pipeline with Silva database version 138.1 (November 2020) [61]. Downstream analyses and graphical outputs were generated with different packages in R version 4.1.2. Krona was used to visualize the abundance of microbial taxonomic composition [62]. Phyloseq R package was utilized to import, analyze, and merge metadata with the raw ASVs counts from the dada2 pipeline [63]. Outliers were detected with PCoA analysis with a 95% confidence interval ellipse inferred using multivariate t-distribution.

Shannon and Simpson’s indices were calculated for alpha diversity estimations. We conducted the Kruskal–Wallis test, followed by subsequent pairwise comparisons with the Wilcoxon rank-sum test comparing alpha diversity metrics among groups (ZIKV infected and non-infected groups; lambda–cyhalothrin-susceptible, moderately resistant, high resistant groups). We assessed community structure (beta diversity analysis) with R package vegan (Community Ecology Package) using the Jaccard distance matrix derived from the OTU matrix [64]. To study the effect of ZIKV infection and lambda–cyhalothrin resistance on microbiota composition variability between samples, we used permutational multivariate analysis of variance (PERMANOVA) with Adonis2 function with 100,000 permutations.

To look for differentially abundant taxa among the different operational taxonomic units (OTUs) among the groups, ZIKV-infected vs. non-infected and lambda–cyhalothrin resistance vs. susceptible characterization was carried out with the Linear Discriminant Analysis Effect Size (LEfSe) algorithm with effect size threshold of 3, and alpha set to 0.05. Other parameters set to default using Microeco R package (version 0.6.5) [65]. In addition, differentially abundant OTUs were also identified for ZIKV-infected vs. non-infected and lambda–cyhalothrin resistance vs. susceptible groups using the ANCOM-BC R package (version 1.4.2) [66]. A Venn diagram was created with groupings for ZIKV and lambda–cyhalothrin resistance.

### 2.6. Prediction of Functional Genes

The software PICRUSt2 (version 2.4.1-beta) [67] with “picrust2_pipeline.py” script was used with default settings to infer the approximate functional potential of the microbial communities associated with the midgut of Colombian *Ae. aegypti* females in ZIKV natural infection and lambda–cyhalothrin resistance profile. We predicted metagenomic diversity from 16S reads with PICRUSt2 and annotated the output with Kyoto Encyclopedia of Genes and Genomes (KEGG) Brite descriptions. Functional predictions were parsed and analyzed for differential abundance with ANCOM-BC [66].

## 3. Results

### 3.1. Natural ZIKV Infection in Colombian Ae. aegypti Females

A total of 62 female mosquitos were analyzed for the presence of ZIKV, 17 of which were positive and represented 27.4% of the total. There were significant differences in the infection status between cities, with a Chi-square (*Χ^2^*): 12.253 df: 5 (*p* < 0.05) (Figure 2). The populations with the highest proportions of infected females were Honda and Bello, with 63.6% and 38.5%, respectively (Figure 2). Neiva, Cucuta, Puerto Bogota, and Acacias presented a percentage of ZIKV infection of 20%, 11.1%, 11.1%, and 10%, respectively (Figure 2).

### 3.2. Bacterial Diversity in Colombian Ae. aegypti Populations According to Natural ZIKV Infection and lambda–cyhalothrin Resistance Status

After verifying the data distribution by PCoA analysis, three samples were classified as outliers and removed from further analyses (Figure 3A). The distribution of the remaining 62 samples was homogeneous (Figure 3B). Later fastq processing, we obtained a total of 1,497,115 reads from 62 female mosquito midguts determined by sequencing analysis of the 16S rRNA gene V3-V4 region on the MiSeq Illumina platform; of these, 1,070,125 reads were from non-infected samples and 426,990 reads were from ZIKV-infected samples.

Bacterial diversity and richness were assessed using the Shannon and Simpson indices, which determine alpha diversity. Although the median of the ZIKV-infected group was lower than that of the uninfected group, there were no significant differences (*p* > 0.05) in species diversity in each community for both indices (Appendix A).

Concerning lambda–cyhalothrin resistance status, two alpha diversity analyses were performed: one grouping all resistant populations (Figure 4A,B) and another with two categories in the resistant phenotype, representing moderate (Puerto Bogotá) and high resistance (Honda, Cucuta, Neiva, and Acacias) (Figure 4C,D). For the Shannon index, all resistant groups had a lower median than the susceptible ones, but there were no significant differences (*p* > 0.05) (Figure 4A,C). The Simpson index was significantly lower (*p* < 0.05) in the lambda–cyhalothrin-resistant midguts in relation to the susceptible profile (Figure 4B) and the moderately resistant group to the other two (Figure 4D).

Using the Adonis2 test, we observed a significant difference in beta diversity for ZIKV infection (*p* = 0.0096, *R^2^* = 0.0666) but not for lambda–cyhalothrin resistance status (*p* = 0.6113, *R^2^* = 0.0242) (Table 1 and Table 2).

Permutational multivariate analysis of variance (PERMANOVA) showed that when infection and resistance are analyzed together, ZIKV infection affected microbiota composition variability based on beta diversity distance matrices (Table 3).

### 3.3. The Bacterial Signature Associated with Natural ZIKV Infection in Colombian Ae. Aegypti Midguts

In *Ae. Aegypti* midguts, 186 OTUs (56.7%) were shared between ZIKV-infected and non-infected females (Figure 5A). We want to highlight that 24 OTUs (7.3%) were unique to natural ZIKV infection (Figure 5A). To further analyze bacterial structure and composition, we used LefSe and ANCOM-BC analyses and found taxonomic groups changed under the condition of ZIKV infection (Figure 5B,C).

The class Bacteroidia (phylum: Bacteroidota) represented 82% of the bacteria associated with ZIKV-infected *Ae. Aegypti* midguts. The family Bacteroidaceae and the species *Bacteroides vulgatus* were taxa that were significantly enriched in ZIKV infection (Figure 5B). At the same time, the phyla Proteobacteria and Firmicutes and the class Gammaproteobacteria were significantly represented in non-infected midguts (Figure 5B). Moreover, genera that decreased significantly under Zika infection were: *Porphyromonas*, *Paraprevotella,* and *Flavobacterium* (class Bacteroidia); *Clostridium*, *Butyricicoccus*, *Pseudoflavonifractor,* and *Blautia* (class Clostridia); *Weissella* (class Bacilli); *Fusobacterium* (class Fusobacteriia); *Megasphaera* (class Negativicutes); *Dorea formicigenerans* (class Clostridia) was significantly increased in ZIKV infection (Figure 5C).

### 3.4. Bacterial Signature in Ae. aegypti Midguts Associated with lambda–cyhalothrin Susceptibility and High and Moderate Resistance

We found that 56.7% (186) OTUs were shared between lambda–cyhalothrin-resistant and susceptible mosquitoes, and 33.2% (109) OTUs were unique for resistant females (Figure 6A). 

The taxonomic group associated with high resistance was the phylum Proteobacteria and for moderate resistance it was *Bacteroides vulgatus* (Figure 6B). In general, the resistant group was enriched significantly with the family *Bacteriodaceae* (Figure 6C) and the genera *Coprococcus* and *Ruminococcus* (class Clostridia); *Bilophila* (class Deltaproteobacteria); *Enterobacter* (class Gammaproteobacteria); *Porphyromonas* (class Bacteroidia); *Bifidobacterium* (class Actinobacteria); *Weissella* (class Bacilli); *Delftia* (class Betaproteobacteria) (Figure 6D). Bacteria of the family *Lachnospiraceae* and the species *Bacteroides faecichinchillae* were significantly decreased in resistant midguts (Figure 6D).

In addition, the midguts from insecticide-susceptible mosquitos were associated with the phyla Actinobacteriota, Firmicutes, and Verrucomicrobiota and the families *Lachnospiraceae* and *Akkermansiaceae*, the latter represented by *Akkermansia muciniphila* (Figure 6B,C).

### 3.5. Prediction of Functional Metabolic Profiling on Bacterial KEGG Genes Associated with Ae. aegypti Midgut Naturally Infected with ZIKV

Bacterial metabolic functions associated with natural ZIKV infection in *Ae. aegypti* midguts were predicted based on 16S rRNA gene sequences and were annotated on KEGGs using PICRUSt2. We found the abundance of 98 KEGGs changed significantly (*p* < 0.05) in ZIKV-infected midguts: 88 decreased and ten increased (Appendix A). These KEGGs were mainly related to metabolic pathways, biosynthesis of secondary metabolites, microbial metabolism in diverse environments, two-component system, ABC transporters, quorum sensing, biofilm formation, pentose phosphate pathway, purine metabolism, glycerolipid metabolism, glycolysis/gluconeogenesis, pentose and glucuronate interconversions, and pyruvate metabolism (Appendix A).

Notably, when the prediction of PICRUSt2 was completed, we found KEGG genes to be associated significantly (*p* < 0.05) with diverse bacterial taxa both in the ZIKV-infected and non-infected conditions (Figure 7). Some of the associations between bacterial taxa and KEGGs in ZIKV-infected midguts were: *Bacteroides* with putative transposase and putative NAD(P)H nitroreductase; *Enterococcus* with polyhomeotic-like protein (PHC1, EDR1) and formylglycine-generating enzyme (SUMF1, FGE); *Pseudomonas* with insecticidal toxin complex protein TccC (tccC), TetR/AcrR family transcriptional regulator–mexCD-oprJ operon repressor (nfxB), membrane protein GlpM (glpM), multidrug efflux pump (mexD), outer membrane protein–multidrug efflux system (oprJ) and ATP-dependent Lon protease (PRSS15); *Bradyrhizobium* with putative transposase, D-ribulokinase and virginiamycin B lyase (vgb); *Campylobacter* with competence proteins (comB4, comB8, and comB9); *Angelakisella* with tccC; *Sutterella* with cytochrome subunit of sulfide dehydrogenase (fccA); *Achromobacter* with a two-component system—OmpR family–sensor kinase ParS (ParS) (Figure 7).

Meanwhile, in intestines not infected with ZIKV, the main associations between KEGGs and bacterial taxa were: *Serratia* with O-antigen polymerase (wzy), LuxR family transcriptional regulator–quorum-sensing system regulator ExpR (expR), aspartate beta-hydroxylase (ASPH), sigma-54 dependent transcriptional regulator–dga operon transcriptional activator (dgaR) and 2-amino-4-deoxychorismate synthase (phzE); *Blautia* with a two-component system comprising the OmpR family and abacitracin resistance response regulator BceR (bceR), as well as a CRISPR-associated protein Csx10 (csx10); *Comamonas* with MSHA pilin protein MshA (mshA); *Asaia* with D-ribulokinase and aminopeptidase; *Acinetobacter* with long-chain-acyl-CoA dehydrogenase (ACADL); *Streptococcus* with a two-component system—LytTR family–response regulator ComE (comE); *Aeromonas* with a two-component system—sensor histidine kinase ChiS (chiS) and the cytochrome subunit of sulfide dehydrogenase (fccA) (Figure 7).

### 3.6. Prediction of Functional Metabolic Profiling on Bacterial KEGG Genes Associated with lambda–cyhalothrin Susceptibility and Resistance in Ae. aegypti

In lambda–cyhalothrin-resistant midguts, 88 KEGGs had a significant change in their abundance; 4 and 84 decreased and increased, respectively (Appendix A). Under this condition, KEGGs were linked principally to metabolic pathways, microbial metabolism in diverse environments, a two-component system, ABC transporters, biosynthesis of secondary metabolites, fructose, mannose metabolism, and methane metabolism (Appendix A).

In parallel, *Pseudomonas* was a genus with more KEGG function associations with lambda–cyhalothrin-resistant females. This genera was related with tryptophan 2-monooxygenase (iaaM); two-component system, OmpR family, sensor histidine kinase PfeS (pfeS, pirS); tccC; two-component system, OmpR family, response regulator PfeR (pfeR, pirR); alpha-1,3-rhamnosyltransferase (wapR); surface adhesion protein (lapA); nfxB; mannuronan synthase (alg44, alg8); alginate production protein (algE); alginate O-acetyltransferase complex protein AlgF (algF); mannuronan 5-epimerase (algG); alginate biosynthesis protein AlgK (algK); alginate biosynthesis protein AlgX (algX); LysR family transcriptional regulator; mexEF-oprN operon transcriptional activator (mexT). Moreover, *Serratia* was related to spermidine dehydrogenase (spdH), fluoroquinolone resistance protein (qnr), and (R)-amidase (ramA) (Figure 8).

## 4. Discussion

### 4.1. Evidence of Natural ZIKV Infection in Rural and Urban Ae. aegypti Populations in Colombia with Different Degrees of Resistance to lambda–cyhalothrin

Similar to DENV infection, the information on ZIKV infection in rural Colombia is very limited [68]. This work provides evidence of ZIKV infection rates for Honda and Puerto Bogotá, two municipalities with less than 24,000 inhabitants in the center of the country, as well as other cities with an epidemiologic history for DENV in Colombia [52]. In our study, Honda, a rural municipality, had the highest ZIKV infection rate in *Ae. aegypti* females (63.6%), corresponding to the ZIKV epidemic in the country between 2015-2017 [15,16,17]. This high rate of arbovirus infection in mosquitoes from sparsely populated places has also been observed for DENV, with a 62% infection rate in rural areas such as Anapoima and La Mesa, Cundinamarca department [68].

The lack of access to regular piped water and its inadequate storage is the main problem and risk factor for the breeding of *Ae. aegypti* in Colombian rural zones [69,70]. Vector control strategies in Colombia are focused mainly in densely populated urban areas; however, evaluating mosquito populations from rural areas is important because vectors with insecticide resistance and high infectivity may be mobilized from them to other areas [68,69,70]. For example, *Ae. Aegypti* specimens collected in Honda presented a high ZIKV infection rate and were highly resistant to lambda–cyhalothrin [47]. Honda is a municipality in the Tolima department with 23,616 inhabitants, where population flow associated with tourism, agriculture, and fishing is common [71]. A high degree of ZIKV transmissibility was reported in Colombia during the 2016 outbreak for municipalities with low population density but high tourist flows [72]. During the ZIKV epidemic in Colombia, Tolima was one of the departments with many cases [16,17].

Meanwhile, the municipality of Bello, with 481,901 inhabitants, is joined to Medellín and is part of the second most populated urban area in Colombia [71]. Females from Bello presented the second largest levels of ZIKV infection (38.5%) in this study and are susceptible to lambda–cyhalothrin [47]. This arbovirus was also detected in *Ae. aegypti* populations from Medellín between 2017 and 2018 with rates of 2.5% to 36.9% [10,73]. Although a low or even null incidence of ZIKV for the Americas [74] and Colombia has been reported [75,76,77,78] during recent years (2019 to 2022), epidemiological surveillance must continue because of the uncertainty of the interaction within epidemic cycles with other arboviruses such as DENV and CHIKV [13,79,80].

### 4.2. Bacterial Signature in Ae. aegypti Midgut Related to Natural ZIKV Infection and Resistance to lambda–cyhalothrin

Alpha diversity summarizes the diversity of microbial communities within a particular condition (ZIKV-infected or non-infected groups; lambda–cyhalothrin-susceptible, moderately resistant, or highly resistant groups) and is usually expressed by the number of taxonomic groups (i.e., species richness) in that ecosystem [81,82]. This work showed quantitatively modulating bacterial diversity; insecticide resistance significantly decreased bacterial richness. Similar to our findings, a reduced diversity of microbial communities was observed in fenitrothion-resistant *Anopheles albimanus* [83]. Meanwhile, beta diversity describes the difference in microbial communities between conditions (comparison between groups) [81,82]. Our results show that the microbial structure in ZIKV-infected midguts differed from non-infected ones. Other studies in *Ae. aegypti* and murine models have shown that ZIKV infection modifies the structure of gut bacterial communities, decreasing their diversity and richness [25,81,84]. Interestingly, our PERMANOVA analysis with both factors, infection and resistance, showed only infection influenced bacterial diversity; this analysis is novel for *Ae. aegypti* because it considers both elements together.

We propose that the lambda–cyhalothrin resistance condition reduces diversity in *Ae. aegypti* populations shape microbial communities at local scales (alpha diversity) and could be a risk factor in the susceptibility of *Ae. aegypti* females to ZIKV infection. This phenomenon of loss of diversity that favors an infection or disease condition has been described in the human gut microbiota [85,86]. Dysbiosis alters the relationships of the existing bacterial communities; diversity is negatively impacted, and unhealthy shifts in bacterial community composition can lower the gut′s resistance to colonization by pathogens [85,87]. This gut dysbiosis has been described in honeybees, where both chronic exposure to low concentrations of insecticides and infection by *Nosema ceranae,* a microsporidian pathogen, may affect the honeybee holobiont [87,88,89,90]. Furthermore, changes in gut bacterial communities by antibiotic treatment in *Culicoides nubeculosus* were associated with a significant increase in infection rate by the Schmallenberg virus [91]. Given this approach, future research should address whether infection by an arbovirus such as ZIKV is favored in a vector with a microbiota shaped by its resistance status. This, in turn, generates different effects on bacterial relationships in the vector’s gut.

The genus *Bacteroides,* belonging to the phylum Bacteroidota, family *Bacteroidaceae*, was predominant in Colombian *Ae. aegypti* midguts in a state of infection with ZIKV and resistance to lambda–cyhalothrin. This phylum was reported in females and males of *Ae. aegypti*, *Ae. albopictus*, *Anopheles spp.,* and *Culex nigripalpus* [44,92,93,94], and the *Bacteroidaceae* family were found in both ZIKV-infected and non-infected *Ae. aegypti* midguts [25]. Furthermore, an unclassified genus from the phylum Bacteroidota, order Bacteroidales, was the most abundant in insecticide-resistant populations of the brown planthopper *Nilaparvata lugens* [95]. Studies in *Ae. aegypti* with experimental ZIKV infection have found associated bacterial taxa such as the families *Rhodobacteraceae* and *Desulfuromonadaceae* and the genera *Cellulosimicrobium, Elizabethkingia, Enterobacter, Pseudomonas,* and *Staphylococcus*; in *Ae. albopictus*, *Elizabethkingia anophelis* was associated with ZIKV infection [25,84,96]. Additionally, *E. anophelis* and *Lysinibacillus* reduced ZIKV infection, affecting vector competence [84,96].

Enteric bacteria such as *Bacteroides* promote virus entry to intestinal epithelial cells in human gut infections [97,98]. Viruses can bind to HBGA (histo-blood group antigens)-like substances, sialylated gangliosides, and lipopolysaccharides (LPS) in bacterial pili and membranes, facilitating virus entry and infection development; an example is the interaction of Norovirus with intestinal bacteria in B cell infection [97,98,99].

Interestingly, the *Lachnospiraceae* family was associated with the lambda–cyhalothrin susceptible Bello population, and *Dorea formicigenerans*, a species belonging to this family, was increased in ZIKV infected midguts. These bacteria have been shown to degrade mucins since they have enzymes involved in sialic acid metabolism, the initial step in the sequential degradation of the mucin oligosaccharide chains, which can serve as a source of energy, carbon, and nitrogen for bacteria [100,101,102]. Moreover, *Akkermansia municiphila*, also associated with the Bello population, was identified as a mucin degrader [103].

Mucins are the main structural components of the insect peritrophic matrix. Although mosquito mucin composition remains largely unknown, it plays a vital role in maintaining homeostasis in the gut by acting as a barrier against invading pathogens and preventing contact with external toxic substances [104,105,106]. Mucin degradation and its impact on infection susceptibility has been documented in humans and mosquitoes [24,107]. For example, *Serratia marcescens*, a gut commensal bacterium, promotes mosquito permissiveness to arboviruses through the digestion of membrane-bound mucins in *Ae. aegypti* gut [24]. Likewise, the Bel protein produced by *Bacillus thuringiensis*, used as a biopesticide for insect control in agriculture, digests intestinal mucin and perforates the midgut peritrophic matrix of *Helicoverpa armigera* larvae, enhancing the toxicity of the Cry1Ac toxin by allowing it to reach gut epithelial cells [108].

We want to highlight that populations highly resistant to lambda–cyhalothrin had a midgut bacterial microbiota associated with the phylum Proteobacteria using the ANCOM-BC differential abundance analysis. We found increased abundance for three genera belonging to this phylum: *Bilophila*, *Enterobacter,* and *Delftia*. Interestingly, this phylum was associated with ZIKV-uninfected females. Our previous studies showed that species belonging to this phylum, mainly *Pseudomonas* and *Rhizobium*, had increased abundance in populations naturally exposed and with a high profile of resistance to lambda–cyhalothrin [47]. Works similar to ours have associated taxonomic groups belonging to the phylum Proteobacteria with insecticide-resistant populations such as *Stenotrophomonas*, *Ochrobactrum*, and *Alphaproteobacteria* in *An. coluzzii* [109], *Xanthomonadales* and *Pseudomonadales* in *Plutella xylostella* [110] and *Burkholderia* in *Riptortus pedestris* [111].

The current study revealed that stressors such as resistance to lambda–cyhalothrin and ZIKV infection could impact vector transmission dynamics by shaping gut bacterial communities. Their influence may be more significant when both occur together in nature.

### 4.3. Predicted Bacterial KEGG Pathways Related to Natural ZIKV Infection and Insecticide Resistance 

Unlike the foregut and hindgut, the mosquito midgut does not have a protective cuticle covering. It is the site of digestion and absorption and the main gut region interacting with environmental contents. Thus, it is the principal target for viruses and their gateway into the hemocoel to establish systemic infection. It is also a microbial niche providing a source of nutrients for bacteria [112,113,114]. Bacteria in the midgut have functions such as modulation of the host immune response, regulating permissiveness or protection from a viral infection, and producing metabolites [23,28,30,40,41,112,115].

Using PISCRUT2 technology, we can predict potential interactions between the vector, midgut bacteria, and the pathogen. Understanding these complex interactions can help explain the different mosquito phenotypes found in the wild. In the present study, the functional predictions were associated with ZIKV infection and non-infection. We found two critical functions by *Bacteroides* associated with ZIKV infection and non-infection. The first were mechanisms for exchanging genetic information, which have been implicated in pathogenicity and the transfer of antimicrobial resistance genes [116]. The second function relates to reductive pathways to degrade or transform polynitroaromatic compounds by bacterial nitroreductases. These flavoenzymes catalyze the NAD(P)H-dependent reduction of the nitro groups on nitroaromatic and nitroheterocyclic compounds such as the pesticides parathion and dinoseb; these enzymes are essential in bioremediation [117,118]. 

Three of the functions predicted for *Serratia* are relevant to vector protection from ZIKV: lipopolysaccharide (LPS) synthesis, quorum sensing, and phenazine biosynthesis [119,120,121,122,123]; all were associated with non-infection. First, LPS synthesis is essential because the O-antigen, one of its significant components highly variable in structure [124], can mediate vector-bacteria interaction, promoting adherence and the ability to overcome host defense mechanisms [120,121]. Second, quorum sensing is a mechanism by which cell–cell signaling controls population density and synchronizes bacterial behavior and social interaction; the vital transcriptional regulators for this function are LuxR family factors, which influence bacterial survival and propagation [122,123]. *Serratia* secretes compounds related to the control of pathogens through suppressed growth, survival, or replication of viruses, bacteria, fungi, and protozoa and that promote the induction of systematic resistance in plants [119,125,126,127]. Finally, we found a pathway related to the synthesis of phenazines, natural bacterial antibiotics that suppress competitors in polymicrobial environments and protect the host against diseases [128,129]. Interestingly, all these pathways described for *Serratia* were decreased in ZIKV infection.

*Asaia* has been found in the midgut of *Ae. aegypti* and *Anopheles spp*. [130,131]. This bacteria can control *Plasmodium* infection in *An. stephensi,* activating the basal level of mosquito immunity [132]. Additionally, it was reported that it impedes *Wolbachia* proliferation in *Anopheles* spp. [133,134]. In our work, this genus had a link with aminopeptidases that decreased significantly in ZIKV infection; these enzymes have been described in *Chromobacterium*, where they inhibit vector infection by damaging the DENV envelope E protein [115].

We previously related *Pseudomonas* with lambda–cyhalothrin resistance. This work confirmed this relationship because *Pseudomonas* had the most KEGG functions associated with this phenotype [47]. We propose that metabolic pathways from *Pseudomonas* may converge and be related to insecticide resistance and susceptibility to infection, opening the opportunity to explore the dysbiosis phenomenon in *Ae. aegypti*. We want to highlight that this genus functions in tryptophan metabolism increased in resistant mosquitoes. It has already been reported that tryptophan catabolism has a role in modulating insecticide tolerance in permethrin-treated *Drosophila melanogaster* [135]. Likewise, this pathway was described in *Anopheles*, where catabolism of 3-hydroxykynurenine (3-HK), a metabolite of tryptophan by *Pseudomonas*, had a role in vectorial competence, protecting mosquitoes against *Plasmodium* infection [136]. ATP-binding cassette (ABC) transporters regulated by a two-component system (TCSs) were another predicted function in *Pseudomonas*. Together, they perform a central role in vital self-immunity, antibiotic transport and resistance processes, and the formation of detoxification modules against AMPs [137,138,139]. ABC transporters also defend against permethrin in the malaria vector *Anopheles stephensi* [140].

Another interesting *Pseudomonas* KEGG in our study, associated with lambda–cyhalothrin-resistant, susceptible, ZIKV-infected and non-infected status is the insecticidal toxin complex protein (tccC). We also found this KEGG in *Angelakisella* associated exclusively with ZIKV infection. Some tccC from bacteria such as *Xenorhabdus* and *Photorhabdus* cause insect death, but others do it at deficient levels or not at all [141,142,143]. Some of these bacterial toxins induce immunosuppression in insects [144,145]; future studies should elucidate the role of tccC produced by *Pseudomonas* in vector competence and insecticide resistance in these *Ae. aegypti* populations.

Another functional prediction with *Pseudomonas* is the TetR-family transcriptional regulators (TFTRs) such as AcrR. These are DNA binding factors that regulate gene expression of efflux pumps and relate to antimicrobial resistance in bacteria [146]. These efflux pumps have been described in *Ae. caspius* as a defense mechanism against insecticides [147].

Finally, another pathway described in our work that we wish to highlight is alginate synthesis related to *Pseudomonas* in resistant *Ae. aegypti* populations. Alginates are produced by bacteria and algae and are used as natural hydrogels for plague control in agriculture. Their function is to contain and release insecticide in a controlled manner preventing environmental damage and affectation of species that are not targets [148,149]. Interestingly, this raises the question of whether these substances may facilitate insecticide encapsulation inside the mosquito, obstructing its action. Future research focusing on alginates from *Pseudomonas* in *Ae. aegypti* should evaluate this hypothesis.

This work contributes knowledge about *Ae. aegypti* populations related to the ZIKV epidemic in Colombia. It will be valuable to plan against future outbreaks and study the impact on other arboviruses such as DENV and CHIKV. We found pathways possibly related to functions such as antiviral, bioremediation, symbiont growth control, pathogenicity, and insecticide resistance. These pathways provide insight into an approximation of natural ZIKV infection, defense against infection and insecticide attack, and susceptibility to pyrethroids. Given this approach, there will be a debate to consider future research that enquires whether a vector with a microbiota shaped by its resistance state can favor a form of infection by an arbovirus such as ZIKV, and this, in turn, can impact bacterial relationships in the vector’s gut.

*Study limitations.* We recognize that the number of samples per city can be considered as limiting. However, the sampling effort to capture females that rest inside the houses using sweep nets yielded a sample size within the expected range (3 to 10 females per sampled locality) [150,151]. The sweep nets method was used because we wanted to capture live females from the field, ensuring a cold chain until they reached the laboratory in order to not modify its microbiota, nor the structure of its scales or body parts. There are some works on microbial diversity of mosquito field populations with bigger samples, but they used adults emerged in the laboratory from larvae or pupae captured in the field, which are more abundant [152,153,154]. However, the manipulation of these immature stages and their cultivation under laboratory conditions, even when they are reared in water from their breeding sites and under sterile conditions, run the risk of modifying their microbiota because of breeding factors such as temperature, water pH, concentration of ions, and food sources [155]. Another limitation of our study is that the PCR used is qualitative and not quantitative; it could be relevant to consider for a future analysis a quantitative PCR that will allow us to correlate the viral load and the respective bacterial species in field mosquitoes.

## 5. Conclusions

This work allowed the identification of bacteria associated with *Ae. aegypti* populations naturally infected with ZIKV, such as *Bacteroides vulgatus*, which was also related to the lambda–cyhalothrin resistance profile. Likewise, we found that the *Lachnospiraceae* family was associated with lambda–cyhalothrin susceptibility, and *Dorea formicigenerans*, a species belonging to this family, was increased in ZIKV-infected midguts. *Bacteroides* were related to pathogenicity, antimicrobial resistance, and bioremediation of insecticide signaling pathways and *Serratia* to regulating the propagation and suppression of competitors. Finally, the role of *Pseudomonas* in resistance to lambda–cyhalothrin with the most associations of defense pathways against insecticides is highlighted. This work provides relevant information on the midgut bacterial microbiota related to resistance and infection and identifies associated metabolic pathways for and against both phenomena. We propose them as potential targets.

## Figures and Tables

**Figure 1 viruses-14-02197-f001:**
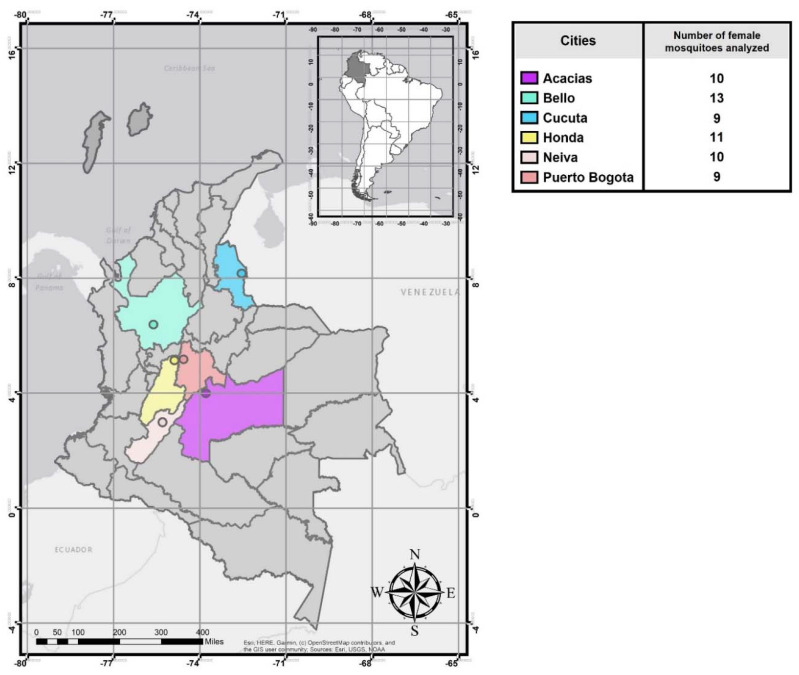
Map of Colombia showing cities where samples of *Aedes aegypti* mosquitoes were collected. The number of female mosquitoes analyzed in each city is indicated in the adjacent table.

**Figure 2 viruses-14-02197-f002:**
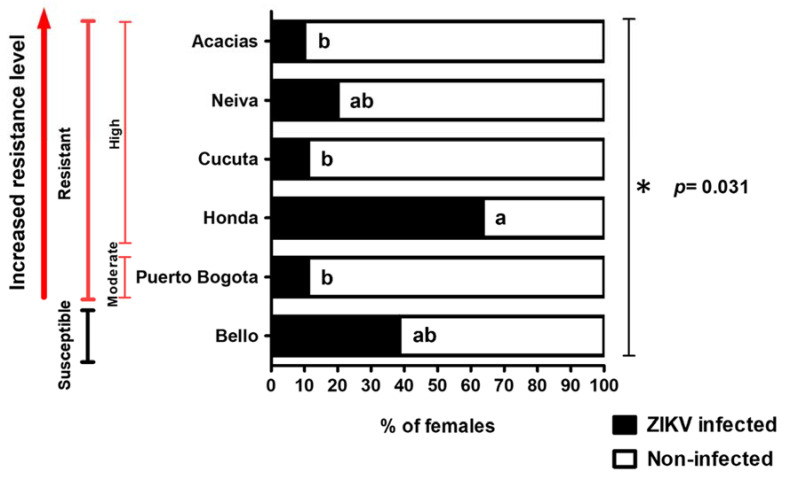
One-step RT-PCR determined the percentage of ZIKV-infected (black) and non-infected (white) females. There were significant differences between cities in infection status. * Chi-square (*Χ^2^*): 12.253 df: 5 (*p* = 0.031). Bars not sharing the same letters are significantly different (*p* < 0.05) in paired comparisons between cities using the Fisher’s exact test.

**Figure 3 viruses-14-02197-f003:**
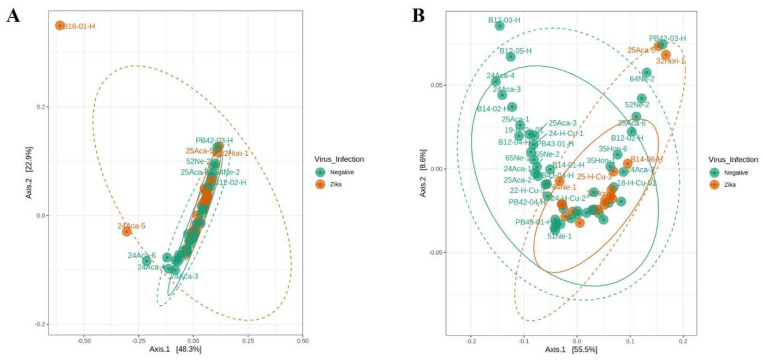
Outlier sample detection by PCoA analysis (Jaccard distance matrix). The analysis was made with a 95% confidence interval ellipse inferred using multivariate t-distribution. Three samples (B16-01-H, 24Aca-5, and 24Aca-6) were outliers and removed from the final analyses. (**A**) Before filtering. (**B**) After filtering.

**Figure 4 viruses-14-02197-f004:**
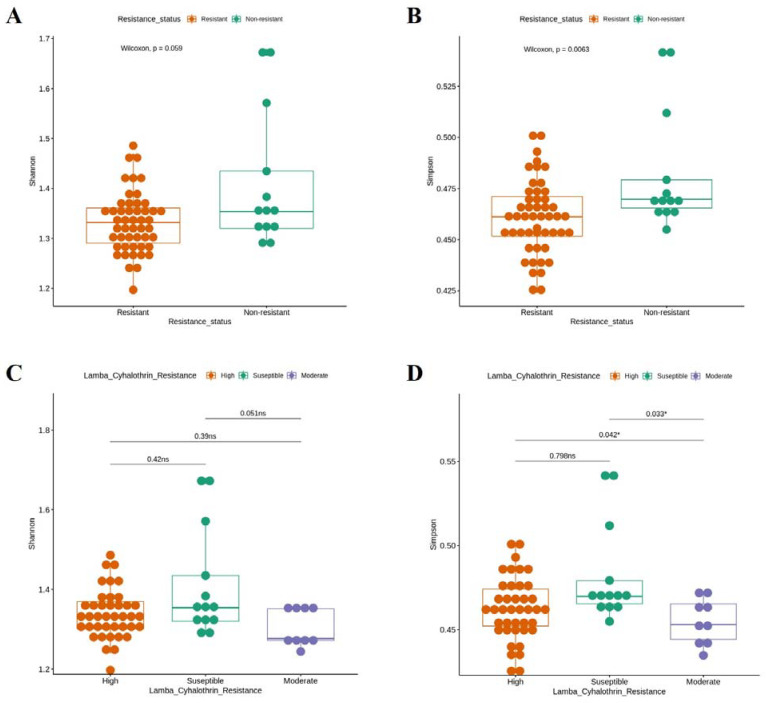
Comparison of bacterial diversity in the midgut of *Ae. aegypti* among resistant and non-resistant (**A**,**B**) and highly lambda–cyhalothrin-resistant, susceptible, and moderately resistant populations (**C**,**D**). Shannon (**A**,**C**) and Simpson (**B**,**D**) alpha diversity indices are shown. Wilcoxon rank-sum tests were conducted under R statistical software version 4.1.2. Groups were significantly different when *p* < 0.05.

**Figure 5 viruses-14-02197-f005:**
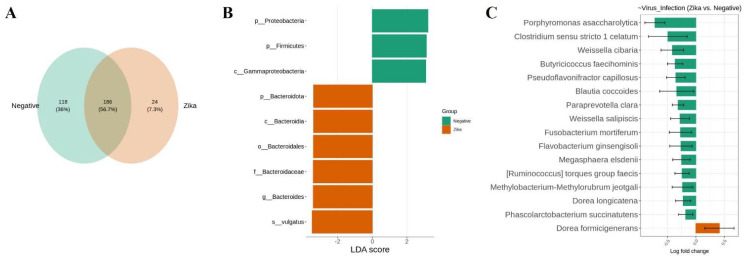
Bacterial community composition of Colombian *Ae. Aegypti* midguts are naturally infected with ZIKV. (**A**) Venn diagram of bacteria showing the common and unique OTUs between ZIKV-infected and non-infected females. (**B**) An abundance of bacterial taxa among groups using LefSe analysis. LDA scores of biomarker bacteria are shown as horizontal bars for biomarker bacteria with an LDA score (log10) > three and with an alpha value of 0.05. (**C**) Differential abundance analysis stratified by ZIKV infection using ANCOM-BC (significant at 5%).

**Figure 6 viruses-14-02197-f006:**
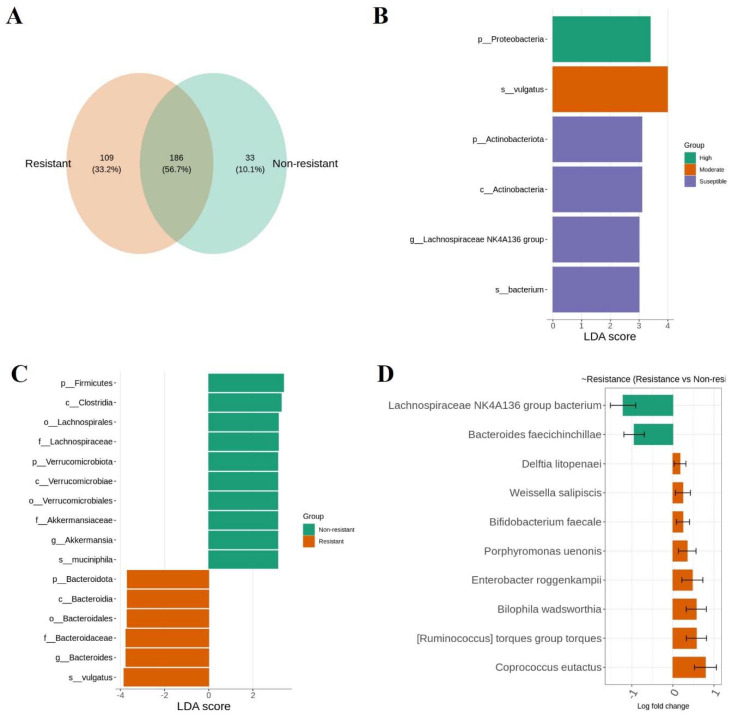
Bacterial community composition of Colombian *Ae. aegypti* midguts associated with lambda–cyhalothrin susceptibility, and moderate and high resistance. (**A**) Venn diagram of bacteria showing the common and unique OTUs in insecticide-resistant and non-resistant females. (**B**,**C**) An abundance of bacterial taxa among groups using LEfSe analysis. LDA scores of biomarker bacteria are shown as horizontal bars for biomarker bacteria with an LDA score (log10) > three and with an alpha value of 0.05. (**D**) Differential abundance analysis stratified by resistance using ANCOM-BC (significant at 5%).

**Figure 7 viruses-14-02197-f007:**
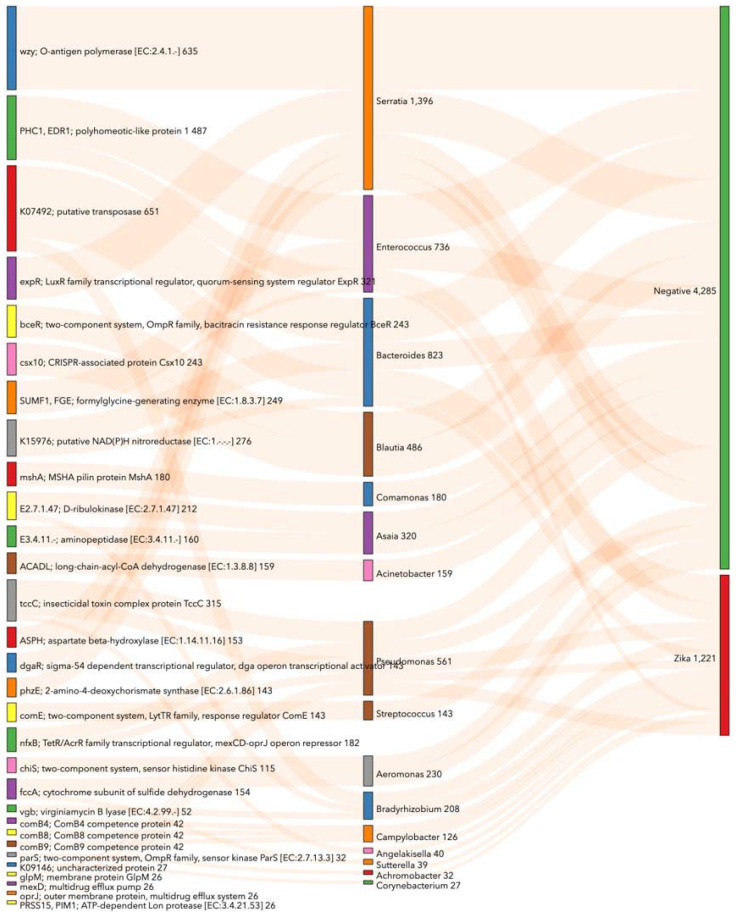
Enriched KEGG genes according to bacteria associated with midgut naturally infected with ZIKV in Colombian *Ae. aegypti*. This Sankey visualizes KEGG pathway groups’ significance (*p* < 0.05) with ANCOM-BC analysis. The numerical value shows PICRUSt2 gene counts corresponding with taxonomy at the genus level. This chart shows the top 20 KEGGs per category by count. For an interactive view, visit https://observablehq.com/d/cd9c60c5c86289ce (accessed on 20 August 2022).

**Figure 8 viruses-14-02197-f008:**
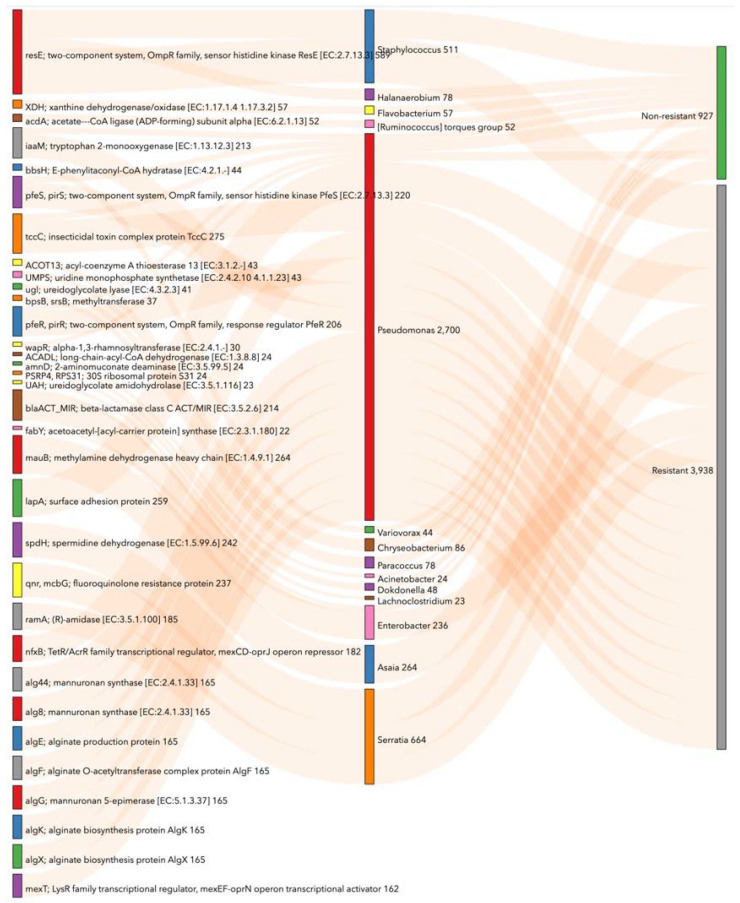
Enriched KEGG genes according to bacteria associated with Colombian *Ae. aegypti* midguts lambda–cyhalothrin resistance. This Sankey visualizes KEGG pathway groups’ significance (*p* < 0.05) with ANCOM-BC analysis. The numerical value shows PICRUSt2 gene counts corresponding with taxonomy at the genus level. This chart shows the top 20 KEGGs per category by count. For an interactive view, visit https://observablehq.com/d/b3e7d8dd95cadd7a (accessed on 20 August 2022).

**Table 1 viruses-14-02197-t001:** Beta diversity analysis of *Ae. aegypti* midgut bacteria for ZIKV infection status using the Adonis2 test (* *p* < 0.05).

	Degrees of Freedom	Sum of Squares	*R^2^*	F	Pr (>F)
**ZIKV infection**	1	0.0483	0.0666	4.2839	0.0096 *
**Residual**	60	0.6768	0.9334		
**Total**	61	0.7252	1.000		

**Table 2 viruses-14-02197-t002:** Beta diversity analysis of *Ae. aegypti* midgut bacteria for lambda–cyhalothrin resistance status using the Adonis2 test.

	Degrees of Freedom	Sum of Squares	*R^2^*	F	Pr (>F)
**lambda-cyhalothrin resistance**	2	0.0175	0.0242	0.7313	0.6113
**Residual**	59	0.7076	0.9758		
**Total**	61	0.7252	1.000		

**Table 3 viruses-14-02197-t003:** Permutational multivariate analysis of variance test (adonis2 function) of midgut bacteria from *Ae. aegypti* populations with different ZIKV infection and lambda–cyhalothrin resistance statuses (* *p* < 0.05).

	Degrees of Freedom	Sum of Squares	*R^2^*	F	Pr (>F)
**Virus infection**	1	0.0528	0.0728	4.7092	0.0063 *
**lambda-cyhalothrin resistance**	1	0.0156	0.0215	1.3882	0.1989
**Residual**	59	0.6613	0.9119		
**Total**	61	0.7252	1.000

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
