# Peer review of "Association of Midgut Bacteria and Their Metabolic Pathways with Zika Infection and Insecticide Resistance in Colombian Aedes aegypti Populations"

_viruses, 2022, doi:10.3390/v14102197_

Round 1

Reviewer 1 Report

This is a well written manuscript with some very interesting data. The authors claim that mosquito gut microbiota changes in response to insecticide resistance and Zika infection. Similar phenomenon have been reported in the literature to support these findings.  The authors make a variety of claims based on data derived from field collected mosquitoes. The sample sizes are small for each location (9-13 mosquitoes) which is not sufficient for a population estimate indicating infection status and location and insecticide resistance. The same can be said for midgut bacteria diversity.

While the methods and analysis are appropriate, the small sample size precludes making population based assertions. The authors should tone down their claims stated in the results and discussion.  The authors should also address the limitations of small sample size on the data.

If possible, could the authors change the orange/red association lines to color with more contrast such as gray, green, or blue in figures 7 and 8?  

Author Response

Point 1: The sample sizes are small for each location (9-13 mosquitoes) which is not sufficient for a population estimate indicating infection status and location and insecticide resistance. The same can be said for midgut bacteria diversity. The authors should also address the limitations of small sample size on the data.

Response 1: We thank the reviewer for this comment and understand his point of view. We have added a section titled “Study limitations” (lines 622-637) in which we recognize the sample size as a limitation and explain the rationale for using the sweep net capture method that yields a low sample size. Additionally, we clarified that the sample size used for evaluation of resistance was not the same as that used to evaluate the microbiota (lines 168 and 184-85).

Point 2: If possible, could the authors change the orange/red association lines to color with more contrast such as gray, green, or blue in figures 7 and 8?

Response 2: We changed the graphic colors as the reviewer suggests, but we believe this change did not favor the contrast for the graphs overall because they include several other colors. For this reason, we have addressed this observation by adding links to a web page where the figures have an interactive format in which the user can highlight the pathways one at a time and obtain a clear view regardless of the line colors (lines 391 and 416).

Reviewer 2 Report

In this manuscript, Andrea Arévalo-Cortés et al. identified bacteria associated with Ae. aegypti populations naturally infected with ZIKV in Colombia, provides relevant information on the midgut bacterial microbiota related to resistance and infection and identifies associated metabolic pathways for and against both phenomena. The paper suggests an interesting natural phenomenon. The manuscript writing is relatively standard. But there are several questions should be addressed.

1. Why was the sampling conducted in a relatively concentrated area?

2. How sensitive is the one-step RT-PCR for the detection of Zika virus? What can be done with a limited sample of mosquitoes to ensure that there are no false negatives?

3. It would be more interesting to correlate the viral loads of different mosquitoes with their microbial species.

Author Response

Point 1: Why was the sampling conducted in a relatively concentrated area?

Response 1: We added an explanation for collection in those cities in the Materials and Methods section (lines 126-136).

Point 2: How sensitive is the one-step RT-PCR for the detection of Zika virus? What can be done with a limited sample of mosquitoes to ensure that there are no false negatives?

Response 2: This RT-PCR has a high level of detection for ZIKV of 140 copies of viral RNA/ reaction [1]. We believe that with this level of sensitivity, false negatives would be rare. This is supported by the fact that previous studies have obtained a similar proportion of positivity (this is mentioned in the first section of the discussion, lines 416-448). Aditionally, we included a positive control (described in lines 157-158) that ensures there are no false negatives for other causes other than low number of viral RNA copies. To ensure no false negatives we would have to implement an additional method, such as isolation and cultivation of the virus. However, this would be beyond our present capacity and the scope of the project, and we would also be limited by the amount of sample.

Point 3: It would be more interesting to correlate the viral loads of different mosquitoes with their microbial species.

Response 3: We agree with the reviewers comment. However, given the qualitative nature of our RT-PCR detection method we currently do not have the capacity to execute this type of analysis. We are considering to establish a quantitative method that will improve our diagnostic capacity. We have added this considerations as limitations to our study in a new section of the Discussion (lines 634-637).

Reference

  1. Balm MND, Lee CK, Lee HK, Chiu L, Koay ESC, Tang JW. A diagnostic polymerase chain reaction assay for Zika virus. Journal of Medical Virology. septiembre de 2012;84(9):1501–5.